# Towards Predictions of Interaction Dynamics between Cereal Aphids and Their Natural Enemies: A Review

**DOI:** 10.3390/insects13050479

**Published:** 2022-05-20

**Authors:** Eric Stell, Helmut Meiss, Françoise Lasserre-Joulin, Olivier Therond

**Affiliations:** 1LAE, Université de Lorraine, INRAE, F-68000 Colmar, France; olivier.therond@inrae.fr; 2LAE, Université de Lorraine, INRAE, F-54000 Nancy, France; helmut.meiss@univ-lorraine.fr (H.M.); francoise.lasserre@univ-lorraine.fr (F.L.-J.)

**Keywords:** agricultural pests, aphids, conservation biological control, models, parasitoids, population dynamics, predator–prey interactions, agroecology

## Abstract

**Simple Summary:**

Understanding how pests and their natural enemies interact dynamically during the growing season and what drivers act on those interactions will help to develop efficient pest control strategies. We reviewed empirical and modeling publications on the drivers influencing the aphids–natural enemy dynamics. We found disparities between what is known empirically and what is used as main drivers in the models. Predation and parasitism are rarely measured empirically but are often represented in models, while plant phenology is supposed to be a strong driver of aphids’ dynamics while it is rarely used in models. Since modelers and empirical scientists do not share a lot of publications, we incite more crossover works between both communities to elaborate (i) new empirical settings based on simulation results and (ii) build more accurate and robust models integrating more key drivers of the aphid dynamics. These models could be integrated into decision support systems to help advisors and farmers to design more effective integrated pest management systems.

**Abstract:**

(1) Although most past studies are based on static analyses of the pest regulation drivers, evidence shows that a greater focus on the temporal dynamics of these interactions is urgently required to develop more efficient strategies. (2) Focusing on aphids, we systematically reviewed (i) empirical knowledge on the drivers influencing the dynamics of aphid–natural enemy interactions and (ii) models developed to simulate temporal or spatio-temporal aphid dynamics. (3) Reviewed studies mainly focus on the abundance dynamics of aphids and their natural enemies, and on aphid population growth rates. The dynamics of parasitism and predation are rarely measured empirically, although it is often represented in models. Temperature is mostly positively correlated with aphid population growth rates. Plant phenology and landscape effects are poorly represented in models. (4) We propose a research agenda to progress towards models and empirical knowledge usable to design effective CBC strategies. We claim that crossover works between empirical and modeling community will help design new empirical settings based on simulation results and build more accurate and robust models integrating more key drivers of aphid dynamics. Such models, turned into decision support systems, are urgently needed by farmers and advisors in order to design effective integrated pest management.

## 1. Introduction

Integrated pest management practices aim to limit chemical pesticides by using complementary tillage and crop management practices (e.g., seed density, planting period) and by promoting conservation biological control (CBC) [1]. CBC consists of spatiotemporally configuring crop and non-crop habitats and adapting agricultural practices to support populations of crop pests’ natural enemies and promote their effectiveness as predators or parasitoids [2,3]. As with integrated pest management practices in general, CBC aims to maintain pest populations below economic thresholds.

Recent studies have increasingly acknowledged the need to account for the dynamic aspects of climate, agricultural landscapes and crop fields when developing CBC strategies [4]. Crop phenology and sequences are strong drivers of insect abundance, diversity, and activity because they determine the spatiotemporal continuity of insects’ resources, both intra- and interannually [5,6,7,8]. The English grain aphid *Sitobion avenae*, for example, can complete its life cycle by dispersing from ripe maize to wheat in autumn [9]. The timing of agricultural practices, e.g., seeding and the application of pesticides, also affect insect dynamics and activities [10], as well as the day-to-day climate [4,11]. These biotic and abiotic drivers of insect populations can lead to unstable effects of CBC by changing the dynamics of pest populations, of their natural enemies, and/or pest–natural enemy interactions [12]. Notions of resonance in the pest–enemy dynamics, such as cyclical oscillations or temporal overlap (synchrony/asynchrony), are crucial to CBC [4,12]. Indeed, for efficient CBC, natural enemies must be present at the same time as the pests they prey upon. Even a small shift in the temporal overlap of the two populations could result in massive growth of the pest population (negative time-delay) or complete pest suppression, hindering the persistence of specialist enemy populations (positive time delay) [7]. Predation and parasitism are expected to have stronger effects if they occur early in the growing season, as demonstrated by theoretical predator–prey models [13,14,15]. Therefore, whereas most studies focus on static CBC approaches, a greater focus on the temporal dynamics of (conservation) biological control is urgently required through empirical and modeling studies [4], see also [16,17,18].

A better understanding of how different biotic and abiotic factors dynamically affect CBC would allow improved predictions of pest population dynamics. Integrating knowledge about these dynamics into models may allow the development of decision support systems for farmers and/or their advisors, for designing effective CBC strategies under the current or future climates [19,20,21]. Furthermore, such models could assess whether pest populations might rise above economic thresholds or not, helping to limit insecticide treatments to strictly necessary situations [22,23].

To further our knowledge on the drivers of CBC dynamics and models supporting adapted CBC strategies, we performed an exhaustive review of the literature dealing with these issues. Due to trophic chain idiosyncrasies [18], we focused our review on crop aphids and their natural enemies in temperate climates. Aphididae are arthropods that act both as phloem-feeders and as a pathogen vectors, transmitting close to 50% of insect-borne viruses to crop plants [24,25]. Aphids include around 5000 species, and about 450 of them are feeding on crop plants, including wheat, soybean, and maize, especially in temperate regions where they are colonizing 25% of plant species [26,27]. Nevertheless, only about 100 species (2%) have exploited the agricultural environment successfully to represent important economic threats [27]. Hill [28] reported that aphids represent today 26% of the 45 major insect pests of the main temperate food crops (maize, wheat, potatoes, sugar beets, barley and tomatoes).

Aphids have a variety of enemies naturally occurring in the ecosystems, including various parasitoid wasps (Hymenoptera-Apocrita), and a variety of predators including coccinellids (Figure 1), lacewings and hover flies but also more generalist natural enemies such as carabids or spiders [29]. Furthermore, aphids are one of the most studied crop pests [18,30]. However, only a fraction of all those studies include the effects of natural enemies. Most of them use only static approaches of these interactions, while temporal or spatio-temporal study are more costly and therefore rarer. We therefore postulated that available studies on biotic and abiotic drivers of aphid–natural enemy interaction dynamics are sufficient to provide added value through their synthesis.

Our goal in this literature review was to identify (i) the abiotic and biotic drivers influencing aphid–natural enemy interaction dynamics and their relative importance (weights); and (ii) the available models to simulate these dynamics that could be used for developing decision support systems. Accordingly, we considered both empirical and numerical modeling studies and discuss the hypotheses used and the knowledge obtained from both corpora. Finally, based on the outcomes of our review, we propose a research agenda to progress towards models and empirical knowledge usable to design effective CBC strategies.

## 2. Materials and Methods

### 2.1. Literature Search

We performed a literature search on the Web of Science Core Collection, considering the 45-year period from 1975 to 20 May 2020, and including only peer-reviewed, English-language articles. We used the wildcard search option (*) to select multiple word endings or plurals. Our final search terms were:

TS=(

(((tempora* OR spati* OR population) NEAR/2 (dynamic* OR variabilit* OR variati*)) OR resilien* OR stabl* OR stabilit* OR robustness) AND (“pest management” OR “biological regulation” OR “biological control” OR biocontrol OR “pest control”) AND (field* OR crop* OR farm* OR agric* OR agro* OR landscape$ OR rotation*) AND (aphid* OR trophic* OR insect$ OR herbivor* OR pest$ OR “natural enem*” OR parasit* OR predat* OR prey*)

NOT (

(resistance AND (expression OR gen*)) OR genom* OR genet* OR *qtl* OR antibio* OR vitamin* OR *marker OR antigen* OR “in vitro” OR recombin* OR bird* OR “house fl*” OR “fruit fl*” OR vineyard* OR orchard* OR “forest management” OR “tropical forest” OR tree* OR ((forest* OR tree*) AND fire*) OR forestry OR “temperate forest*” OR “forest pest”))

Our search returned 1580 publications. All search results were first reviewed at the title level and then at the abstract level to ensure they met the following inclusion criteria:the studied organisms (pests and natural enemies) must be invertebrates;studies must consider annual field crops in temperate climates, and;studies must report temporal and/or spatiotemporal population dynamics (of pests and/or their natural enemies) or pest–natural enemy interaction dynamics (predation, parasitism).

After this two-step manual selection, 343 publications were selected for further consideration. We then focused our literature research on aphids, i.e., as hypothesized, the most frequently studied crop pest group (Appendix A). Of the 343 selected publications, 107 concerned aphids and were retained for our full analysis, in which we distinguish between empirical and modeling studies.

After reading, we made a final selection based on the same inclusion criteria mentioned above. Our final selection yielded 64 empirical publications and 17 modeling publications. Finally, we included six additional modeling studies that were not returned by our Web of Science search, but which we consider to be important, bringing the final modeling corpus to 23 publications. We describe the full conception process of our corpus in Figure 2.

### 2.2. Analysis and Data Assimilation

To perform our analysis, we extracted detailed information from each publication into empirical and modeling databases. For each empirical study, we recorded the country in which the study was performed, the crop considered, the aphid and natural enemy species, the temporal and spatial extents and resolutions, the frequencies of insect sampling, the time steps between samplings and the spatial sampling units. For each modeling study, we recorded the ecological processes included in the models and all the input and output variables.

Moreover, we recorded all explicitly enounced ecological hypotheses and also reported when empirical or modeling studies implicitly tested a pre-identified hypothesis through their results analysis (Appendix A).

In both corpora, we recorded all biotic and abiotic predictors and response indicators studied and all studied relationships between them. All these indicators provide information on temporal or spatiotemporal dynamics, i.e., are indicators of dynamics (vs. static state). When analyzing the relationships between the predictor and response indicators, we indicated a positive relationship when the statistical results of the study showed a significant positive relationship between the predictor and response indicators over time. A negative relationship indicates a significant negative relationship in the study between the predictor and response indicators over time. To reduce ambiguity arising from the vocabulary used in the selected publications, we followed the recommendations of Ratsimba et al. [18] and aggregated indicators (predictors and responses) into clear indicator names (e.g., “aphid abundance” or “temperature”); the ecological hypotheses tested in each study were treated in the same way. Following [18], hereafter, the term “relationship” describes a studied relation between a predictor indicator and a response indicator. A relationship may have been measured once or multiple times in one or more studies. Still following [18], “Measurement” describes a single measurement of a predictor–response relationship performed in one study. In both the empirical and modeling corpora, we counted every measurement and the observed sign of the relationship (positive, negative, or nonsignificant).

Furthermore, we counted the number of publications analyzing more than one predictor for the same response indicator. Then, from those publications, we systematically extracted data ranking the explicative power of the investigated predictors. Based on this relative ranking information, we attempted to order the predictors from the most to the least explicative.

## 3. Results

### 3.1. Overview of Selected Publications

The (spatio-)temporal dynamics of aphids, their natural enemies, and/or their interactions were rarely studied between 1975 and 2003, with no WoS registered publications before 1995 and only about one publication per year between 1995 and 2003. Since then, the number of publications has increased, fluctuating between three and eight publications per year. Of the 87 relevant publications reviewed here, there were many more empirical (*n* = 64) than modeling (23) studies (see Appendix A).

The reviewed publications were performed in 22 countries distributed over six continents, with most coming from the United States (*n* = 27), followed by China (11) and the United Kingdom (9). Studies on aphid–natural enemy dynamics were performed on a large array of crop types, but predominantly wheat (*n* = 31) and soybean (20). Of the various aphid species studied, the three most studied were *Sitobion avenae* (*n* = 23), *Aphis glycines* (20), and *Rhopalosiphum padi*; this is unsurprising because *S. avenae* and *R. padi* are major wheat pests and *A. glycines* are very frequent on soybean plants. Two groups of aphid natural enemies dominate the literature, coccinellid (*n* = 36) and parasitoid wasps (34), whereas other predator groups featured in fewer than 15 publications (see Appendix A).

The entire corpus of 87 articles includes 242 individual authors. Many of these scientists authored only empirical articles (71%), 20% only modeling studies, and only 9% published both types of studies (Figure 3a). About half of the modeling publications (*n* = 11) focused only on aphid dynamics, whereas the rest (10) dealt with aphid interactions with parasitoids (4), predators (5), or both (1; Figure 3b). In contrast, empirical studies focused more on interactions between aphids and their natural enemies (*n* = 55) than on aphids alone (8; Figure 3b).

To study the dynamics of aphids and their natural enemies, empirical studies were most often based on weekly time sampling/observation intervals (*n* = 29), followed by bi-weekly (10), semiweekly (8), and daily (6) intervals; monthly (1) or annually (2) sampling intervals were rarely used (Figure 4). The total number of samplings also showed high variability but tended to increase with decreasing sampling interval. Surprisingly, five publications claiming to study insect dynamics were based on only one or two samplings, which seems to be too infrequent to characterize any dynamic trends. Nine publications did not mention their sampling frequency, and 15 did not report their total number of samplings per year.

### 3.2. Relationships between Predictor and Response Indicators

#### 3.2.1. Quantification in Empirical and Modeling Studies

We identified 252 single measurements of predictor–response relationships reported in the 87 reviewed publications (Table 1). The three most studied response indicators were (i) abundance dynamics of aphids (*n* = 61) (ii) abundance dynamics of their natural enemies (59), and (iii) aphid population growth rates (53). Other key response indicators studied were linked to aphid control over time: parasitism (*n* = 12), predation (11), pest suppression (7), and biocontrol (1). Some relationships also measured the effects on insect migration (*n* = 12), community diversity (9), intraguild predation among natural enemies (2), and populations spatiotemporal stability (7).

In the following two subsections, we analyze in detail the relationships between predictors and the three most often studied response indicators (Table 1): the abundance dynamics of aphids, those of their natural enemies, and aphid population growth rate. Results on other response indicators (predation, parasitism, migration/flux) were not common enough (fewer than five measurements) to draw sound conclusions; we include those results in Appendix A.

#### 3.2.2. Drivers of Aphid Abundance and Population Growth Rates and Their Natural Enemies Abundance Dynamics

We registered 32 different predictor indicators for the three most frequently investigated responses (aphid abundance dynamics, natural enemy abundance dynamics, and aphid population growth rates) and a total of 173 measurements concerning 64 unique predictor–response relationships. Table 2 summarizes these relationships and their number of measurements, distinguishing empirical from modeling studies.

Aphid abundance dynamics were studied through 61 measurements using 25 different predictor indicators, aphid growth rate through 53 measurements using 18 predictor indicators, and natural enemy abundance dynamics through 59 measurements using 21 predictor indicators.

Natural enemy abundance was the most frequently studied predictor of aphid abundance dynamics (*n* = 9) and the second most studied predictor of aphid growth rate (11; after temperature, see below). Seven measurements (six empirical, one modeling) showed this predictor to be negatively correlated with aphid abundance dynamics (at daily or weekly scales), whereas two measurements were nonsignificant at a weekly scale. The same trend was observed when natural enemy abundances were used to predict aphid growth rates, with all (but one) negative measurements at all different time scales. Conversely, aphid abundance was the most frequently studied predictor of natural enemy abundance dynamics, with a majority of positive reported measurements (*n* = 10 on 16) and no negative one, at daily and weekly scales.

Intercropping was the second most often studied predictor of aphid abundance dynamics (*n* = 5), with four negative reported measurements and only one positive. This intercropping effect was always analyzed at weekly timesteps. This predictor was also the second most important for natural enemy abundance dynamics; four measurements showed positive correlations (weekly samplings, 3 to 11 sampling dates) and two showed negative correlations (weekly sampling, 5 to 12 sampling dates).

Concerning climatic factors, temperature was the most often studied predictor of aphid growth rates; the vast majority of reported measurements were positive (*n* = 11, including both empirical and modeling studies), except three nonsignificant empirical measurements, and this was independent of the sampling frequency. The effect of temperature on aphid abundance dynamics is less clear, with only two negative and one nonsignificant reported measurement. In the case of precipitation, only three empirical studies investigated their impact on weekly aphid abundance dynamics with no clear trend over the long period (annual). Two studies found that precipitation does not significantly impact aphid population growth rate.

Landscape complexity often had nonsignificant effects on aphid abundance dynamics (*n* = 3) or natural enemy abundance dynamics (3), either when considering daily or yearly time steps. The percentage of semi-natural habitat in the landscape was reported to positively (*n* = 2) or negatively (2) affect aphid abundance, but was only shown to positively affect natural enemy abundance (3, weekly sampling through season or between years). None of the reviewed studies have analyzed the direct or indirect effects of landscape structure on aphid population growth rates.

Alternative resources for natural enemies (alternative prey or floral resources) do not show a clear trend with natural enemy abundance dynamics; two positive, one negative, and two nonsignificant measurements have been reported.

#### 3.2.3. Relative Rankings of Predictor Indicators in Conjoint Analyses

Of the 39, 33, and 27 publications investigating predictors of aphid abundance dynamics, natural enemy abundance dynamics, and aphid population growth rate, respectively, we found 15, 19, and 12 studies, respectively, which conjointly analyzed the effects of multiple predictor indicators. However, only 11, 15, and 5 of those publications provided information regarding the explicative power (relative importance) of those indicators. This can provide a first step to a ranking of the main drivers of the three investigated response indicators (Appendix A).

Regarding aphid abundance dynamics, relative humidity had higher explicative power than other climatic indicators (precipitations, temperature) [31,32]. Natural enemy abundance was more predictive of aphid abundance dynamics than aphid density [33], but less important than crop type [34] or climatic indicators [31]. The percentage of semi-natural habitat was found to be less explicative of aphid abundance than precipitation [35] or predation [36] but more explicative than crop type [37] (or landscape complexity [38]).

Concerning aphid population growth rates, the time within the season was more explicative than natural enemy abundance [12], which, in turn, was more explicative than aphid abundance [39] or agricultural intensification [40]. Bommarco et al. [41] found aphid abundance to be more explicative of population growth rates than precipitation or temperature, whereas Chen and Hopper [42] found no significant effects of these three predictors.

Relating to natural enemy abundance, we identified three groups of studies: Chaplin-Kramer et al. [43] and Raymond et al. [12] studied the effects of landscape complexity on multiple natural enemies; Elliot [39], Hesler [44] and Rhainds et al. [45] investigated the effects of aphid abundance on coccinellids, *Orius* (minute pirate bugs), and other natural enemies; and Evans [46] and Yoo and O’Neil [47] compared the effects of aphid abundance and natural enemies’ alternative resources on natural enemy abundance. All other predictor comparisons were unique. Predictor rankings must be described according to the different natural enemies present and/or their growth stages, confirming the hypothesis that trophic chain idiosyncrasies dominate pest population dynamics. For example, aphid abundance had a superior impact on coccinellid larvae than on adults, and a lesser impact on syrphids. Except for *Orius*, natural enemies’ alternative resources were less important than aphid abundance for predicting natural enemy abundance. Relative humidity is better than aphid abundance, precipitation, or temperature for predicting coccinellid abundance. The amount of semi-natural habitat in the landscape is more important than landscape complexity for all investigated natural enemies; more precisely, landscape complexity is a good predictor of coccinellid abundance, but not syrphid or carabid abundances. The occurrence of harvesting (agricultural practice) explains well the dynamics of coccinellids but not carabids.

### 3.3. Ecological Hypotheses

Three hypotheses were most often tested regarding the interactions between aphids and their natural enemies. Ten empirical studies validated the hypothesis that the density of natural enemies strongly influences aphid dynamics throughout the growing season [44,48,49]. The hypothesis that biocontrol is enhanced when natural enemies establish populations earlier in the season, so they can stop or significantly reduce aphid colony growth before it became too large, was validated by ten empirical studies [45,50,51], whereas only one study showed variable results [52]. This hypothesis was also included for the development of models in three modeling studies. The hypothesis that natural enemy populations respond spatiotemporally to variations in aphid population density, by following the dynamics of prey patches in the field, was confirmed by eight empirical studies but rejected by three others. Two publications state that there is a synchrony between the peaks of aphid populations and predators’ voracity.

The importance of alternative resources (flowers, alternative preys) as a positive factor to enhance natural enemy populations in early seasons was validated by three studies but rejected by one. Alternative resources can be useful to sustain natural enemy populations and to attract them closer to or into the fields, when aphids, their main diet, will start to colonize it.

Among abiotic factors, four empirical studies validated the temperature as a main factor driving aphid–natural enemy interactions by changing the intrinsic species biological parameters and the synchrony of the different species dynamics, allowing them to interact, whereas two others rejected this hypothesis.

Concerning landscape effects, six publications tested and validated the hypotheses that semi-natural habitats are sources of both natural enemies and aphids and their proximity to field crops leads to a better aphid colonization, but also to an easier natural enemy dispersal and thus an enhanced biocontrol. Two studies worked on the effects of complex landscapes on an earlier effectiveness of aphid predators, but there was no consensus on the results. Four studies also confirmed the hypothesis that aphid population dynamics are temporally linked to plant phenology throughout the growing season because aphid fitness strongly depends on their host plants’ quality for feeding and reproduction.

Finally, four studies validated the hypothesis that aphid population densities early in the season are important for aphid population growth rates throughout the rest of the season, as predators cannot impact aphid colony growth once the population is already too large.

## 4. Discussion

By synthesizing information on peer-reviewed empirical and modeling studies, our review clarified how agronomists, entomologists, ecologists, and modelers are assessing aphid–natural enemy interaction dynamics and the main outcomes of their studies. We identified some consensus concerning the effects of predictor indicators on response indicators and associated ecological hypotheses. We also noted some conflicting results, and overall a great dispersion among the predictor indicators studied.

### 4.1. The Need for Crossover between Empirical and Modeling Studies

Modeling studies often focused on the aphid life cycle and population dynamics (10 of 22 publications), and only a few evaluated interactions between aphids and their natural enemies leading to aphid CBC. In contrast, many empirical studies analyzed aphid–natural enemy interaction dynamics, e.g., by using exclusion cages. Nonetheless, only a few directly measured the action of aphid regulation by their natural enemies (predation, parasitism). Most empirical studies underlined a negative correlation between aphid population dynamics and those of their natural enemies.

These particularities between the two types of studies led us to assume that the outcomes of empirical studies of aphid–natural enemy interaction dynamics have, as yet, rarely been directly used in modeling studies. Moreover, very few publishing scientists span both communities (Figure 3a). These results, as also noted by Petit et al. and Donatelli et al. [20,53], clearly highlight the need for crossover studies to produce efficient and realistic models that integrate empirical findings to predict aphids’ and other pests’ population dynamics based on the influence of their naturally occurring enemies. Such crossover studies would also promote the design of empirical studies to test modeling results, and vice versa [20,54]. As an example, model simulations can help to formulate hypotheses that could then be tested experimentally, in turn providing data necessary to calibrate, validate, and improve the models. Bridging the empirical and modeling communities might be promoted by the development of facilitating tools. Malard et al. [55] developed a generic platform to facilitate the development of modular and predictive food-web models, even for users that are not familiar with numerical simulations and/or coding.

### 4.2. A Diversity of Models but a Lack of Integration of Key Drivers

We found a diversity of models in our corpus. On the one hand, Bahlai et al. [56] presented a purely mechanistic model (with many functions and parameters) depicting the detailed stages of the population dynamics of aphids and their natural enemies, but also of the crop plants. In their model, transitions between developmental stages are triggered by parameter thresholds (e.g., temperature, degree-days, chronological time). Such detailed models integrate a large panel of information on the trophic chains. On the other hand, simplified models describe aphid dynamics throughout the cultural season with simple population logistic growth equations that include only a few parameters such as aphid population growth rate, temperature, mortality sources (predation, parasitism), or crop carrying capacity [57,58,59]. Curtsdotter et al. [60] developed a research-oriented food-web model that can integrate an unlimited number of predators and alternative prey species in the food-web. By using an allometric hypothesis for predator–prey interactions, this model drastically reduces the number of parameters to estimate, making it easier to instantiate.

The aphid life cycle during the crop season (including reproduction and mortality) appears to be well-known, and is, and has been for a long time, frequently integrated in models [61]. In contrast, processes occurring outside of the crop season are less well known and poorly represented in models. Only two of the 23 reviewed modeling studies integrated aphid overwintering or diapause. Similarly, spatial aphid dispersal processes (immigration and emigration) were only represented in three models. Although the effects of landscape characteristics are frequently emphasized in the overall CBC literature (see Introduction), there seems to be a scarcity of models simulating spatio-temporal aphid movements across agricultural landscapes during the growing season and from year to year.

Surprisingly, predation and/or parasitism processes are not often investigated in the empirical corpus, but are often represented in models. Many models represent predation or parasitism dynamics as Type-II or Type-III functional responses [57,62]. Other works recognize the scarcity of available information on certain parameters, such as parasitoid attack rates on aphids, and use their models to explore those parameters’ influences on aphid dynamics by varying their value over realistic ranges [63].

In general, the temporal phenology of the host plant is poorly represented in models while it can be a strong driver of aphid arrival and population (de-)growth [56,60]. In addition, the importance of alternative prey and alternative plant resources for natural enemies is a well-supported ecological hypothesis. It has been shown to strongly affect the dynamics of natural enemies, as well as indirectly influencing aphid abundance and population growth rates [64,65]. Unfortunately, no modeling publication in our corpus integrated alternative plant resources to explore their effects on aphid and natural enemy populations, and only Curtsdotter et al. [60] integrated alternative prey and intraguild predation in their food-web model. This may be due to the difficulty in representing alternative resources and their interaction dynamics with natural enemies. Therefore, most models represent only aphid dynamics.

The influences of landscape structure on aphid and natural enemy dynamics are now well-known to be highly contextual [17]. Indeed, in our corpus, we obtained inconsistent responses for all landscape predictors. The effects of landscape structure on aphid regulation dynamics were only investigated by Bianchi et al. [58], who modeled the effects of semi-natural habitats on natural enemy abundance. As with alternative resources, using models to explore the effects of landscape structure on aphid and natural enemy dynamics should be a major objective in agroecological research. Such modeling studies might help to decipher how landscape structures drive spatio-temporal insect dispersal from field to field, crop to crop, alternative hosts, or overwintering sites [7].

Finally, although aphid damage to plants, pest harmfulness thresholds and the temporal variation of plant sensitivity are key factors that should be taken into account to design low- or no-pesticide cropping systems, we found no dynamic models dealing with them.

Many models of our corpus were intended to explore only one driver of aphid population dynamics and no model integrated all the key parameters identified in this review. Integrating more drivers and more complex food webs comes with the need for estimating more input parameters. This trade-off was alleviated by Curtsdotter et al. [60] who integrated a large number of food-web interactions including, alternative prey and intraguild predation by proposing an allometric modeling approach, reducing the number of parameters describing the feeding interactions from 294 to only five. Empirical knowledge on landscape complexity effect on aphid and natural enemy dynamics [12] should be integrated in models, and static models such as the one by Jonsson et al. [66] might be transformed to dynamic models.

### 4.3. Key Drivers of Aphid Dynamics and Research Issues

Our study provides a first step in the attempt to rank the different drivers of the dynamics of the abundances of aphids and their natural enemies. This ranking is based on separate publications that conjointly analyzed different drivers and may not be generalizable as contexts may be different. However, it does provide a first list of key factors that should be jointly investigated in future experiments to decipher their relative importance.

Empirical and modeling studies on the dynamics of parasitism or predation processes itself and the factors driving their dynamics remain scarce, reflecting the difficulty in measuring and discriminating the various causes of aphid mortality. Nevertheless, development of new emerging technologies, such as Next Generation Sequencing (NGS) or DNA-based molecular gut content analyses in predators’ guts [67,68], will improve our understanding of food-web structures and trophic interactions. However, the use of those methods remains complex and expensive. Despite these difficulties, some strong relationships have emerged from our analysis of both empirical and modeling corpora. Studies agree that high natural enemy abundances reduce aphid abundance dynamics and population growth rates. This relationship constitutes a robust consensus in our corpus. Additionally, ten publications validated the hypothesis that natural enemies influence aphid dynamics over the growing season. Thus, by extension, we can assume that the abundance of natural enemies constitutes a good proxy of aphid CBC over the season [33,44,48], although the natural enemy species or even functional group differ between situations. However, it is important to note that climatic factors [31] or crop type [34] are often more important drivers of aphid population dynamics when directly compared to natural enemy abundance dynamics.

Whereas ten publications supported the hypothesis that the early establishment of natural enemies enhances aphid CBC, temporal indicators describing the emergence or arrival of natural enemies and aphids are rarely used. These indicators are very important and should be standardized for future studies. We also suggest that early-season measurements of natural enemy and aphid emergence or arrivals and early population dynamics should be a more systematic target for future CBC studies. Several studies [41,45,69,70] validated the hypothesis that initial aphid population size is a good predictor of aphid population dynamics during the crop growth cycle, affecting aphid population growth. The initial conditions of the pest–natural enemy system thus appear to be strong drivers of population dynamics and should be systematically and precisely investigated using standardized indicators, which still need to be defined.

In accordance with the reviews of Letourneau et al. [71] and Petit et al. [20], our review confirms that intercropping may have a major positive impact on CBC as all relevant publications in our corpus validated this ecological hypothesis. Intercropping may offer supplementary trophic resources and shelters to natural enemies, promoting their populations (top-down aphid control) [65]. It could also act as a resource (host plant) dilution that reduces pest infestation or as a physical or chemical disturbance for herbivores (bottom-up control) [20]; for instance, garlic volatiles have been shown to repulse aphids [72].

Many studies have observed positive correlations between air temperatures and aphid growth rates through the season [61], with only a few reporting nonsignificant relations [41,42,73]. Temperature was also the most frequently included variable in modeling studies (10 of 22 publications). Temperature is either included in the population growth equation [57] or as a threshold parameter conditioning the evolution of a modeled population [56]. High temperature can also have a negative effect on aphid dynamics, leading to suboptimal growth. While the temperature effect is more often studied, the ranking of predictors suggests that humidity affects aphid and coccinellid dynamics even more strongly than temperature, but this finding is only based on two studies [31,32]. Nonetheless, the combined effects of relative humidity, precipitation, and temperature should be investigated further.

In their review, Ratsimba et al. [18] stated that most relationships between landscape drivers and CBC have only been investigated by one or very few studies due to the high variability of predictor and response indicators used in the research community. The same conclusion can be drawn in our study regarding the investigated relationships and ecological hypotheses, strongly limiting the possibility of drawing general conclusions. Landscape ecological hypotheses such as “the proximity between fields and SNH enhance biocontrol” or “in complex landscapes, predators are more effective in the early season” were only tested in three and two studies, respectively, with variable results, while the usefulness of SNH and complex landscapes in enhancing biocontrol is often validated by static studies [6,74,75,76], but see Tscharntke et al. [77] and Karp et al. [17]. Surprisingly, very few landscape predictors have been investigated in the studies in our corpora. The few studies show rather weak or nonsignificant correlations, sometimes in opposite directions, that have never been ranked as important predictors. In contrast, several studies have validated the effects of semi-natural habitats on aphid colonization rate [35,70,78]. To clarify these effects, as highlighted by Ratsimba et al. [18] and Holland et al. [79], future studies should investigate the effects of different semi-natural habitats according to the specific trophic chain studied.

## 5. Conclusions

Identifying robust drivers of the spatio-temporal aphid dynamics is the most promising way to build actionable knowledge, such as model-based decision support systems, to develop efficient conservation biological control programs against aphids, one of the most important crop pests in temperate climates. Through this review, we participate to identify and synthetize knowledge on these drivers and highlight the need for further research focusing on the dynamics of the studied system rather than describing only static relationships as classically done.

By systematically reviewing both empirical and modeling studies of the population dynamics of aphids and their natural enemies, we highlighted existing consensuses and controversies in both research communities. Existing studies focused on three key components of CBC: aphid abundance dynamics, aphid population growth rates, and their natural enemies’ abundance dynamics. We synthesized information on the identified drivers (predictor indicators) of these dynamics and, when possible, their relative importance. Reviewed studies mostly validated three key ecological hypotheses concerning aphid–natural enemy relationships. Natural enemy abundance is a robust negative predictor of aphid abundance dynamics, and, conversely, aphid abundance is a robust positive predictor of natural enemy abundance dynamics. However, humidity and temperature seem to be even more important predictors of aphid abundance dynamics.

Among the research reviewed here, only few publications and few authors bridge the empirical and modeling communities. We encourage crossover studies that may facilitate (a) the development of more realistic models and (b) the design of experimental settings based on testable hypotheses issued from simulations. We highlighted the main strengths, weaknesses and missing points of existing models and identified key modeling strategies for their improvements such as non-modeler-oriented modeling architectures or allometric hypothesis-based models strongly reducing the number of required parameters. These latter may allow to adapt these models to different situations with variable insect communities. This is necessary to provide agricultural actors with operational decision support systems that they can be used to design effective integrated pest management based on ecosystem services. Predicting natural regulation could make it possible to partially dispense with systematic and very costly field observations.

## Figures and Tables

**Figure 1 insects-13-00479-f001:**
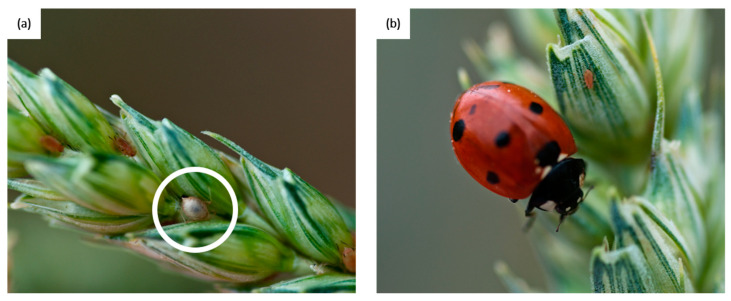
Aphids (here *Sitobion avenae*) are one of the most studied pests of cereal crops. Their population dynamics are potentially reduced by their natural enemies: (**a**) aphids mummies (white ring) are the result of parasitism, and (**b**) coccinellids are one of the most important aphid’s predators. Photos by Véronique Tosser.

**Figure 2 insects-13-00479-f002:**
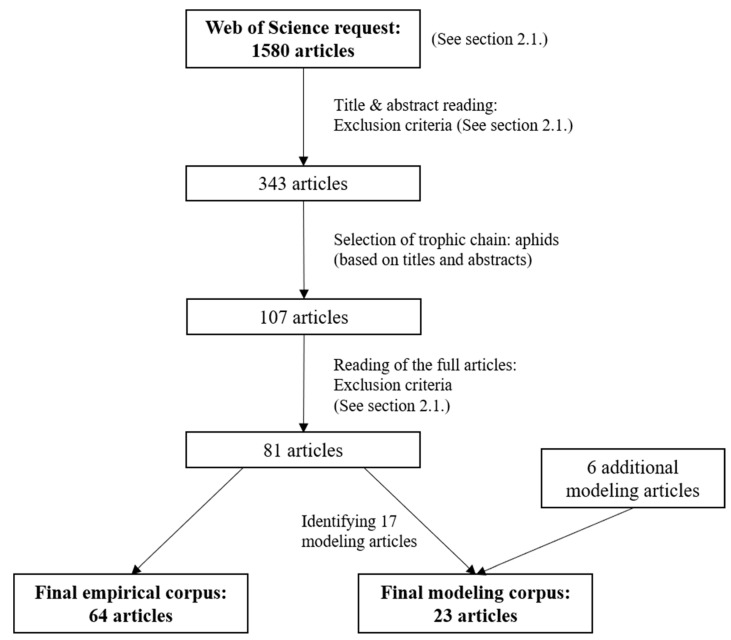
Flow diagram of the corpus conception from the request in WebOfSciences to the final corpus.

**Figure 3 insects-13-00479-f003:**
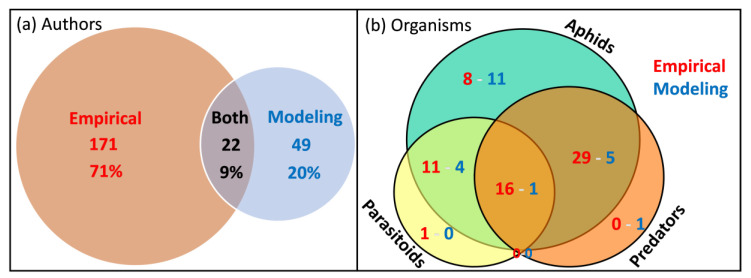
(**a**) Number of authors (scientists) in the entire corpus that co-authored the 64 empirical and/or the 23 modeling studies. Only 22 scientists (9%) are authors of both types of studies. (**b**) Numbers of empirical and modeling publications studying aphids, predators, or parasitoids alone, as well as 2- or 3-way combinations. For example, 16 empirical studies but only one modeling study conjointly analyzed all three functional groups of organisms.

**Figure 4 insects-13-00479-f004:**
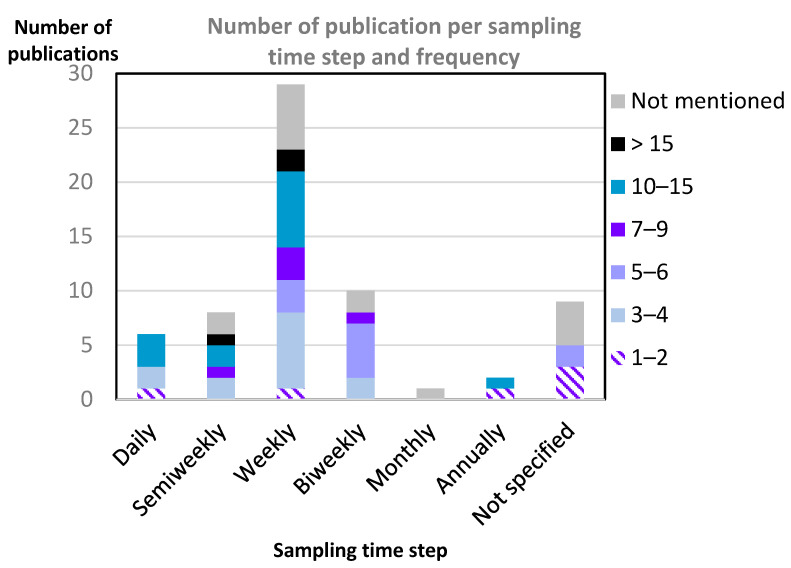
Sampling frequency (time between successive samplings, x-axis) and the total number of annual samplings (colors) in the 64 empirical studies.

**Table 1 insects-13-00479-t001:** Numbers of measurements of predictor–response relationships, sorted by response indicators (first column) and the functional group concerned by the response indicator (natural enemies, aphids, or neither), in the 87 selected empirical and modeling studies. Response indicators are always considered through their evolution in time.

	Natural Enemies		Aphids		Crop Outcomes	Total
Response Indicators	Empirical	Modeling	Empirical	Modeling	Empirical	
Abundance	54	5	52	9	-	120
Population growth rate	7	6	44	9	-	66
Migration/flux	1	-	8	3	-	12
Parasitism	-	-	10	2	-	12
Predation	-	-	8	3	-	11
Community diversity	7	-	2	-	-	9
Agricultural results *	-	-	2	-	6	8
Pest suppression	-	2	4	-	-	6
Spatiotemporal stability	3	-	2	-	-	5
Intraguild predation	-	-	-	2	-	2
Biocontrol	-	-	-	-	1	1
Total	74	13	132	26	7	252

* Agricultural results: crop damage (*n* = 2), yield (5), and field above economical threshold (1).

**Table 2 insects-13-00479-t002:** The number of published positive (Pos), negative (Neg), and nonsignificant (NS) measurements between predictor variables (rows) and three response variables (three main columns). Results are sorted by predictor variable category and distinguish between empirical (E) and modeling studies (M). SNH = Semi natural habitat.

	Response Variable																		
	Aphids Abundance Dynamics						Aphids Growth Rate Dynamics						Enemy Abundance Dynamics						TOT
Predictor Variable	Pos		NS		Neg		Pos		NS		Neg		Pos		NS		Neg		
	E	M	E	M	E	M	E	M	E	M	E	M	E	M	E	M	E	M	
Insecticide use	-	-	-	-	2	-	-	-	-	-	4	-	-	-	1	-	2	-	9
Fertilizer use	-	-	-	-	-	1	-	-	-	-	-	1	-	-	-	-	-	1	3
Tillage	-	-	-	-	1	-	-	-	-	-	-	-	-	-	1	-	1	-	3
Sowing date	1	-	-	-	-	-	-	-	-	-	-	-	-	-	-	-	-	-	1
Insecticide delay	-	-	-	-	-	-	-	-	-	-	-	-	1	-	-	-	-	-	1
Temperature	1	-	-	-	1	1	6	5	3	-	-	-	-	-	1	-	-	-	18
Precipitation	1	-	1	-	1	1	-	-	2	-	-	-	1	-	-	-	-	-	7
Humidity	-	-	-	-	2	-	-	-	1	-	-	-	3	-	-	-	-	-	6
Atmospheric CO_2_	-	1	-	-	-	2	-	1	-	-	-	1	-	-	-	-	-	-	5
Wind speed	-	-	-	-	-	-	-	-	1	-	-	-	-	-	-	-	-	-	1
% intercropping	1	-	-	-	4	-	-	-	-	-	1	-	4	-	-	-	2	-	12
Intensification	1	-	1	-	-	-	-	-	1	-	-	-	2	-	-	-	1	-	6
Crop type	1	1	-	-	1	-	-	-	-	-	1	-	-	-	1	-	-	-	5
Agroforestry	-	-	2	-	-	-	1	-	-	-	-	-	-	-	1	-	-	-	4
% natural borders	-	-	-	-	1	-	-	-	-	-	-	-	-	-	-	-	-	-	1
Irrigation	1	-	-	-	-	-	-	-	-	-	-	-	-	-	-	-	-	-	1
Aphid abundance	2	-	-	-	-	-	-	-	2	-	3	-	9	1	6	-	-	-	23
Enemy abundance	-	-	2	-	6	1	-	-	1	-	10	-	-	-	-	-	-	-	20
Alternative resources	-	-	-	-	2	-	1	-	-	-	-	-	2	-	2	-	1	-	8
Enemy diversity	1	-	-	-	-	-	-	-	-	-	-	-	-	-	-	-	-	-	1
Predation	-	-	-	-	1	-	-	-	-	-	-	-	-	-	-	-	-	-	1
Migration / flux	-	-	-	-	-	-	1	-	-	-	-	-	-	-	-	-	-	-	1
Parasitism	-	-	-	-	-	-	-	-	-	-	1	-	-	-	-	-	-	-	1
Aphid growth rate	-	-	-	-	-	-	-	-	-	-	-	-	-	1	-	-	-	-	1
Landscape complexity	1	-	3	-	-	-	-	-	-	-	-	-	1	-	3	-	2	-	10
% SNH	2	-	-	-	2	-	-	-	-	-	-	-	2	1	-	-	-	-	7
% grassland	1	-	-	-	1	-	-	-	-	-	-	-	-	-	-	-	-	-	2
% crop	-	-	-	-	-	-	-	-	-	-	-	-	-	1	-	-	-	-	1
SNH proximity	-	-	-	-	-	-	-	-	-	-	-	-	-	-	1	-	-	-	1
Timing in season	1	-	-	-	-	-	1	-	-	-	-	-	1	-	-	-	-	-	3
Plant stage	2	-	-	-	1	-	1	-	1	-	2	-	1	-	-	-	-	-	8
Plant morphology	-	-	1	-	-	-	-	-	-	-	-	-	1	-	-	-	-	-	2
TOTAL =	61						53						59						173

## Data Availability

Not applicable.

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
