# Peer review of "Towards Predictions of Interaction Dynamics between Cereal Aphids and Their Natural Enemies: A Review"

_insects, 2022, doi:10.3390/insects13050479_

Round 1

Reviewer 1 Report

The manuscript is interesting and writing is fine.  Just some sections should be improved 

1)  boolean expression issues

 I think this is bigger Boolean expression.

This computer (such as database for reference) cannot work well for your search target.

This might influence your searching and scientific finding

I strongly suggest authors can separate the different items for looking for the key or useful references 

And this should be supported as the supplementary files 

2) suggest figure 2 can be merged in this supplementary files

3) Table 2 should be improved.

Details can be seen in the attachment file.

Author Response

Dear reviewer,

Reviewer 2 Report

Dear Authors,

The research presents many interesting aspects and food for thought. Basically, it reflects the point of view of researchers who know little about the insects in question (aphids and their natural antagonists); but, this could also be a positive fact: the point of view of "non-specialists" can give a vision more impartial and objective. So, that's okay. However, I strongly invite you to take my suggestions that I inserted in the pdf into account and learn more about this group of insects.

Reviewer 3 Report

In the article “Towards predictions of interaction dynamics between cereal aphids and their natural enemies: A review”, the authors aim to revise the empirical and modelling works on these relevant pests and their biological control by natural enemies with a special focus on temporal dynamics. While I think that the topic is highly important and that a synthesis of the large number of studies focused on aphids on annual crops is needed, after reading this draft I believe that some more work is needed to provide future readers with a clear synthesis on the topic. Please see my detailed comments below and some minor observations in the end. Sorry if my comments sound too harsh, but I think that the authors have done a good job at identifying a set of studies on the topic and that some more work can be done to improve the depth of their interpretation to deliver a paper that will be useful to many readers.

Major observations

-I believe that some articles could be omitted in the literature search given that the search terms did not include anything on aphids specifically. I understand that focusing on pests, aphids would be a subset of these studies, but why not include them directly in the search terms? It seems the most straightforward option to me. For example, by making a short search I found that studies such as Schmidt et al (2005) and Thies et al (2005) are missing.

-My feeling after reading the results and discussion sections is that the review fell short in interpreting the results of the studies and it is mainly a count of the different drivers and the number of studies that investigated them. Even if the first part of the results section is needed to describe the publications, I think that perhaps too many figures and text are dedicated to this and the discussion seems shallow. The section on the different drivers shares some of these issues, as the text describes what can be read in that very large table, whereas the rankings of drivers are based on separate articles and might not be so generalizable. Finally, the section on the different hypotheses seems like the most interesting one and could be extended a bit to include short descriptions of the hypotheses in the text, for example.

-While the review claims to be focused on dynamics, most of the text is not clearly pointing to that, or at least it is hard to see it. Perhaps the authors could re-organize the results to highlight those studies focused on spatial and temporal dynamics separately, and within the temporal dynamics try to show examples at the different scales that were considered.

-Finally, I don’t think that English writing should be a reason to reject an article unless it is an extreme case. This article is well-written in most of its parts, but I did find several small issues that complicate its reading. For example, check the first four minor observations, which are all within the first lines of the article. Please revise your text in detail for these small things in your revised version.

Minor observations

-L 8: *understanding

-L 9: the growing season? Cultural might not be so clear for all readers

-L 13: but are often

-L 15: do not share

-L 50: please check that all scientific names are written in italics

-L 81: I believe that in cases like this you should cite the author/s surname to start the sentence.

-L 83-84: this last sentence is a bit repetitive to what was said earlier in the paragraph. Perhaps it can be omitted or rephrased?

-Table 1: not clear why the column is called “not attributed”. If all the studies in that column are linked to effects on crops, the column could be called Crop outcomes, crop performance, or simply crop. Also, the sum of that column says 7 but only 6 studies are listed above.

-Table 2: how should the effects of the planting date be interpreted? A positive effect on aphids implies that late sowing led to higher aphid abundances?

Also, this table in general seems too large and difficult to read due to size. For example, in the lower rows, it is not simple to interpret if the number belongs to an empirical or a modelling study, and the direction of the effect. I can’t think of an easy way to improve this, but perhaps the authors can rethink the way this information is given. Since a lot of this information, or at least the most important trends, are described in the text perhaps it is not necessary to show everything in the main text.

-Supplementary materials S2: check the sub-heading on page 1, it says S1

-L 285-287: this sentence is not so clear, please rephrase.

References cited:

-Schmidt, M. H., Thewes, U., Thies, C., & Tscharntke, T. (2004). Aphid suppression by natural enemies in mulched cereals. Entomologia Experimentalis et Applicata113(2), 87-93.

- Thies, C., Roschewitz, I., & Tscharntke, T. (2005). The landscape context of cereal aphid–parasitoid interactions. Proceedings of the Royal Society B: Biological Sciences272(1559), 203-210.

Round 2

Reviewer 1 Report

This vesion is much better than before. I can say it can be accpeted from this presentation.

Author Response

Dear reviewer,

We would like to thank you for your precious feedbacks.

Reviewer 3 Report

Thank you for your revision of the manuscript. I am satisfied with the answers provided by the authors and the changes that were made in the text, which I think is clearer now and acknowledges some limitations of the review when needed. In my opinion, this is much closer to being ready for publication, and I only have some minor comments listed below (the line numbers refer to the manuscript with tracked changes).

-L 97: are sufficient?

-L 100: cereal crops.

-L 159: and the spatial sampling units.

-L 264: their evolution in time?

-L 307: “and this independently of” this part of the sentence is not so clear. Perhaps “and this was independent of the sampling frequency?

-L 380: state.

-L 408-409: indicators?

-L 476: “Therefore,”.

-L 533: you repeated “that” by mistake.

-L 537: which still need to be defined.

-Some parts of the discussion are fragmented into too many short paragraphs. Perhaps some of them could be merged, such as those focusing on temperature and humidity (lines 548-562) or those related to landscape effects (lines 563-580). In my opinion, something similar happens in the conclusions, where paragraphs from lines 590 to 601 could be merged.

-L 558: based.

Author Response

Dear reviewer,

We would like to thank you for your final feedbacks that really helps to come up with a better manuscript. We follow your final suggestions for minor revisions.

Thank you for your revision of the manuscript. I am satisfied with the answers provided by the authors and the changes that were made in the text, which I think is clearer now and acknowledges some limitations of the review when needed. In my opinion, this is much closer to being ready for publication, and I only have some minor comments listed below (the line numbers refer to the manuscript with tracked changes).

Minor Comments:

*          L 97: are sufficient?

Ok.

*          L 100: cereal crops.

Ok.

*          L 159: and the spatial sampling units.

Ok.

*          L 264: their evolution in time?

Ok.

*          L 307: “and this independently of” this part of the sentence is not so clear. Perhaps “and this was independent of the sampling frequency?

Ok.

*          L 380: state.

Ok.

*          L 408-409: indicators?

Ok.

*          L 476: “Therefore,”.

Ok.

*          L 533: you repeated “that” by mistake.

Ok.

*          L 537: which still need to be defined.        

Ok.

*          Some parts of the discussion are fragmented into too many short paragraphs. Perhaps some of them could be merged, such as those focusing on temperature and humidity (lines 548-562) or those related to landscape effects (lines 563-580). In my opinion, something similar happens in the conclusions, where paragraphs from lines 590 to 601 could be merged.

Ok.

We also merged paragraphs from lines 507 to 526.

*          L 558: based.

Ok.